

# Auroral ionospheric E region parameters obtained from satellite-based far ultraviolet and ground-based ionosonde observations: Effects of proton precipitation

Harold K. Knight[1]

[1]Computational Physics, Inc., Springfield, VA, 22151, USA

*Correspondence to*: Harold K. Knight (knight@cpi.com)

10 Dec. 2020

**Abstract.** Coincident auroral far ultraviolet (FUV) and ground-based ionosonde observations are compared for the purpose of determining whether auroral FUV remote sensing algorithms that assume pure electron precipitation are biased in the presence of proton precipitation. Auroral particle transport and optical emission models, such as the Boltzmann 3-Constituent (B3C) model, predict that maximum E region electron density (NmE) values derived from auroral Lyman-Birge-Hopfield (LBH) emission assuming electron precipitation will be biased high by up to ~20% for pure proton aurora, while comparisons between LBH radiances and radiances derived from in situ particle flux observations (i.e., *Knight et al.* 2008, 2012) indicate that the bias associated with proton aurora should be much larger. Surprisingly, in the comparisons with ionosonde observations described here, no bias associated with proton aurora is found in FUV-derived auroral NmE, which means that auroral FUV remote sensing methods for NmE are more accurate in the presence of proton precipitation than was suggested in the aforementioned earlier works. Possible explanations for the discrepancy with the earlier results are discussed.

## 1 Introduction

Auroral particle transport and optical emission models, such as the Boltzmann 3-Constituent (B3C, *Strickland et al.* [1993]) model, predict that proton aurora is ~50% more efficient in producing $N_2$ Lyman-Birge-Hopfield (LBH) emission than electron aurora, but comparisons between satellite-based auroral far ultraviolet (FUV) and in situ particle flux observations have indicated that the difference is actually much greater than 50% [*Knight et al.*, 2008, 2012]. Such a bias would have a detrimental effect on auroral FUV remote sensing algorithms that assume pure electron precipitation. In this study, we look for a biasing effect of proton precipitation on the auroral maximum ionospheric E region electron density parameter, NmE, derived from LBH radiances, by comparing FUV observations with coincident observations of the E region by high latitude ground-based ionosondes. Supporting information for this study is given in a recent paper [*Knight et al.*, 2018], including details about instruments, our auroral FUV remote sensing algorithm (assuming pure electron precipitation), and ionogram interpretation methods. *Knight et al.* [2018] also gives extensive comparison results for the coincident FUV and ionosonde observations, but omits an investigation of the statistical effect of proton aurora, which is left for this paper.

Although proton aurora contributes only ~15% of the total energy deposited by auroral precipitation into the upper atmosphere [*Hardy et al.*, 1989], it is often an important source of E region ionization, especially towards the equatorward boundary of the auroral oval in the pre-midnight sector [*Galand and Richmond*, 2001]. *Galand and Lummerzheim* [2004] pointed out that the different ion production and emission profiles of electron and proton aurora will cause errors in LBH-based auroral remote sensing methods that assume electron precipitation. Besides the two *Knight et al.* studies mentioned above, *Frey et al.* [2001], *Gérard et al.* [2001], and *Coumans et al.* [2002] also compared FUV and in situ observations. These earlier works are discussed in *Knight et al.* [2008].

In studying possible biases associated with proton aurora, one advantage of comparing FUV data with E region observations instead of in situ particle flux is that E region comparisons eliminate the uncertainty associated with possible differences in calibration errors between electron and ion (treated as proton) flux detectors. We opted to use ionosonde instead of incoherent scatter radar (ISR) observations of the E region because ionosondes operate continuously and offer thousands of observations that are made within a few minutes of satellite overpasses of ground stations. We have made auroral FUV-

ionosonde comparisons for approximately 1000 overpasses of ground stations, which is larger than any previous study of coincident FUV and ground-based observations of the E region.

FUV observations from three instruments aboard three different satellites were included in the study: the NASA Thermosphere, Ionosphere, Mesosphere Energetics and Dynamics (TIMED) Global Ultraviolet Imager (GUVI, *Christensen et al.* [2003]) and the Defense Meteorological Satellite Program (DMSP) F16 and F18 Special Sensor Ultraviolet

Spectrographic Imager (SSUSI, *Paxton et al.* [2002]). Section 2.1 reviews information on these instruments given in *Knight et al.* [2018] and also gives extra information relevant for proton auroral emissions. Section 2.2 gives a very brief summary of Digisonde [*Reinisch*, 1996; *Reinisch et al.*, 2009] ionosonde data sources and interpretation methods, which are described in detail in *Knight et al.* [2018]. Ionogram analysis for this work was performed by the University of Massachusetts, Lowell, Space Science Laboratory. Coincident FUV images, ionograms, and extracted parameters are provided at the following

website created for this project: http://www.cpi.com/projects/fuvi.html. Section 3 describes the expected effects, given the above-mentioned B3C model and FUV/in situ comparison results, of proton precipitation on the accuracy of auroral FUV remote sensing algorithms that derive E region parameters in terms of LBH emission assuming pure electron precipitation.

Then Section 4.1 presents actual statistical effects of proton aurora found in the coincident FUV and ionosonde NmE observations. Surprisingly, no statistical effect on the accuracy of FUV-derived NmE is found. As described in *Knight et al.*

[2018], strong statistical agreement was found between FUV-derived and ionosonde-observed auroral NmE. In the former, NmE is derived from precipitating particle characteristics inferred from LBH emission. In the latter, NmE is inferred from the maximum radio frequency at which signal echoes are received by ionosondes (roughly speaking). Since both observation methods give information on auroral NmE (albeit, using very different methods), and since a strong statistical relationship was found between the two types of NmE observations in *Knight et al.* [2018], it is to be expected that if proton precipitation

biases FUV-derived NmE then such an effect could be detected statistically in the FUV-ionosonde comparisons. In fact, we quantify the expected statistical effect using a statistical simulation in the appendix. Possible explanations for the unexpected lack of a proton auroral bias in FUV-derive NmE are considered in Section 4.2.

While good statistical agreement between the two types of auroral E region observations was reported for NmE in *Knight et al.* [2018], poor agreement was reported for hmE. Based on statistical analysis (omitted here), it is clear that this lack of

agreement is unrelated to effects of proton precipitation. The reader is referred to *Knight et al.* [2018] for a discussion of possible reasons for the lack of agreement in hmE. Here, we do not include statistical analysis of effect of proton aurora on

the level of agreement in hmE, since it appears that there are other factors involved that prevent such a comparison from giving meaningful results on the effect of proton precipitation on FUV-derived hmE accuracy.

## 2 Data and methods

### 2.1 Auroral FUV

This subsection will mostly be a review of Section 2 of *Knight et al.* [2018], along with some additional information specific to the Ly-α line, i.e., HI 121.6 nm, which is produced by proton aurora. For additional details, the reader is referred to *Knight et al.* [2018]. TIMED is in an orbit that precesses ~0.2 hours of local time per day, while F16 and F18 are in sun-synchronous orbits. Each of the three satellites passes through the northern (as well as the southern) high-latitude region once every ~100 minutes. The three FUV instruments record images of radiances in five FUV channels, referred to as 1216, 1304, 1356, LBHS, and LBHL, the first three corresponding to the HI 121.6 nm, OI 130.4 nm, OI 135.6 nm lines, and the last two corresponding to two wavelength intervals in which $N_2$ LBH radiance predominates. These two channels are referred to as LBH short and LBH long, respectively. Their wavelength intervals are approximately [140 nm, 152 nm] and [165 nm, 180 nm], respectively, except for early F16 SSUSI LBHS, which was a bit different, as described in *Knight et al.* [2018]. When we use the term "1216" without wavelength units, it is meant as an instrument channel name and not as a scientific term for the HI 121.6 nm feature.

LBHS and LBHL responsivities per pixel for the three instruments are given in Table 1 of *Knight et al.* [2018]. The GUVI, F16 SSUSI, and F18 SSUSI 1216 responsivities in counts per kR per pixel, are 1.3, 3.7, and 2.1, respectively. (This applies to calibrations versions C, E, and B, respectively, as in Table 1 of *Knight et al.* [2018].) We extract radiance values for this study by averaging over pixels within a 30 km radius of the location of the coincident ground station for an overpass. At nadir, there are ~57.5 pixels per averaging area for GUVI and ~28.5 for SSUSI. Only half of the SSUSI 1216 pixels values are downlinked by F16 and F18. The missing pixel values are provided in Level 1B (L1B) files by means of an estimate in terms of surrounding pixel values. This gives effective 1216 responsivities of 0.075, 0.053, and 0.030 counts per R per averaging area for GUVI, F16 SSUSI, and F18 SSUSI, respectively.

A total of 2047 qualifying overpasses by the three satellites over the four ground stations were found from for the time period covered by our study, i.e., from 2002 through 2014. For us, a qualifying overpass is one for which SZA is at least 100°, LBHL is at least 200 R, the FUV instrument look angle is less than 40°, and there are no errors in the FUV data. For the analysis described here, there was an additional condition for F16, which was that F16 SSUSI LBHL should be at least 300 R. This condition was added because of the poorer agreement between F16 SSUSI and ionosonde observations described in *Knight et al.* [2018] Section 4.1. This work required the subtraction of geocorona from 1216 radiances to give an estimate of the auroral 1216 signal. We developed an algorithm for fitting geocorona to 1216 radiance images, but the

algorithm could not be applied to all overpasses (e.g., because of missing L1B data). The overpasses for which geocorona could not be estimated are excluded from the analysis described here. Geocorona removal increases the relative uncertainty in the remaining signal as explained in, e.g., *Knight et al.* [2008], paragraph 35.

This study uses the exact same method for deriving NmE and hmE from auroral FUV as is described in *Knight et al.* [2018] Section 2.3. The method assumes that LBHS and LBHL values are produced by pure electron aurora with Gaussian (i.e., nearly monoenergetic) spectra. As illustrated by *Knight et al.* [2018] Fig. 1, NmE and hmE are estimated from LBHS and LBHL using tabulated values indexed only by the ratio LBHS/LBHL and instrument look angle. As described in Section 4.1, the hypothesis that proton aurora causes a bias in auroral FUV algorithms that derive NmE from LBH was tested by

characterizing the statistical effect of auroral HI 121.6 nm (associated with proton precipitation) on biases between FUV-derived and ionosonde-observed NmE.

## 2.2 Ionosonde

Ionosonde observations were provided by four Digisondes located at magnetic latitudes between 60º and 70º: Gakona (AK, USA), Goose Bay (Canada), Norilsk (Russia), and Tromso (Norway). All ionogram-derived data values for this study were

provided by University of Massachusetts, Lowell, Space Science Laboratory. Our method for extracting auroral ionospheric E region parameters from ground-based ionosonde observations is described in detail in Section 3 of *Knight et al.* [2018]. The ionosondes record ionograms once every ~15 minutes, generally, and for every overpass there will be two ionograms to be considered, i.e., immediately before and after the exact time of coincidence. Auroral E region parameters could be extracted from one or both of the ionograms for ~55% of the qualifying overpasses, and in ~30% of these cases, auroral E

region parameters could not be extracted for exactly one of the two ionograms (meaning ~15% for the pre-coincidence ionogram and ~15% for the post-coincidence ionogram).

## 3 Expected effect of proton precipitation

Proton precipitation is expected to make auroral FUV remote sensing algorithms based only on LBH (and assuming pure electron precipitation) inaccurate because proton and electron precipitation produce LBH emission and E region ionization in

different quantities per unit precipitating energy flux. Figure 1 illustrates some of these expected differences. The values shown in the figure were generated with the B3C model, based on the same assumptions as in Section 2.3 of *Knight et al.* [2018] except for one difference, which is that in Fig. 1 the contribution to LBH of cascade from the $N_2(a')$ and $N_2(w)$ states to the $N_2(a)$ state is not included. Instead, all of the LBH cross sections (including electron, proton, and hydrogen atom impact) are as described in *Knight et al.* [2012]. The reason for this is that we do not have cross sections for proton and

hydrogen atom impact that describe cascade at this time and the inclusion of cascade for electrons but not protons and hydrogen atoms would not be self-consistent.

Figure 1 shows B3C results based on nadir viewing for Gaussian electron spectra, in addition to three proton flux kappa function (e.g., *Knight et al.* [2012]) spectra for $\kappa$ values of 3.1, 6.2, and 100. The kappa function becomes wider (as a function of energy) for smaller $\kappa$ values and approaches a Maxwellian shape as $\kappa \to \infty$. The three $\kappa$ values used in the figure are representative of the range of proton spectral shapes that have been observed (e.g., *Coumans et al.* [2002]). The incident flux spectra all have spectrally integrated energy flux (denoted by $Q$) equal to 1 erg cm$^{-2}$ s$^{-1}$. $E_0$ values outside of the ranges shown for the three proton auroral $\kappa$ values do not occur generally.

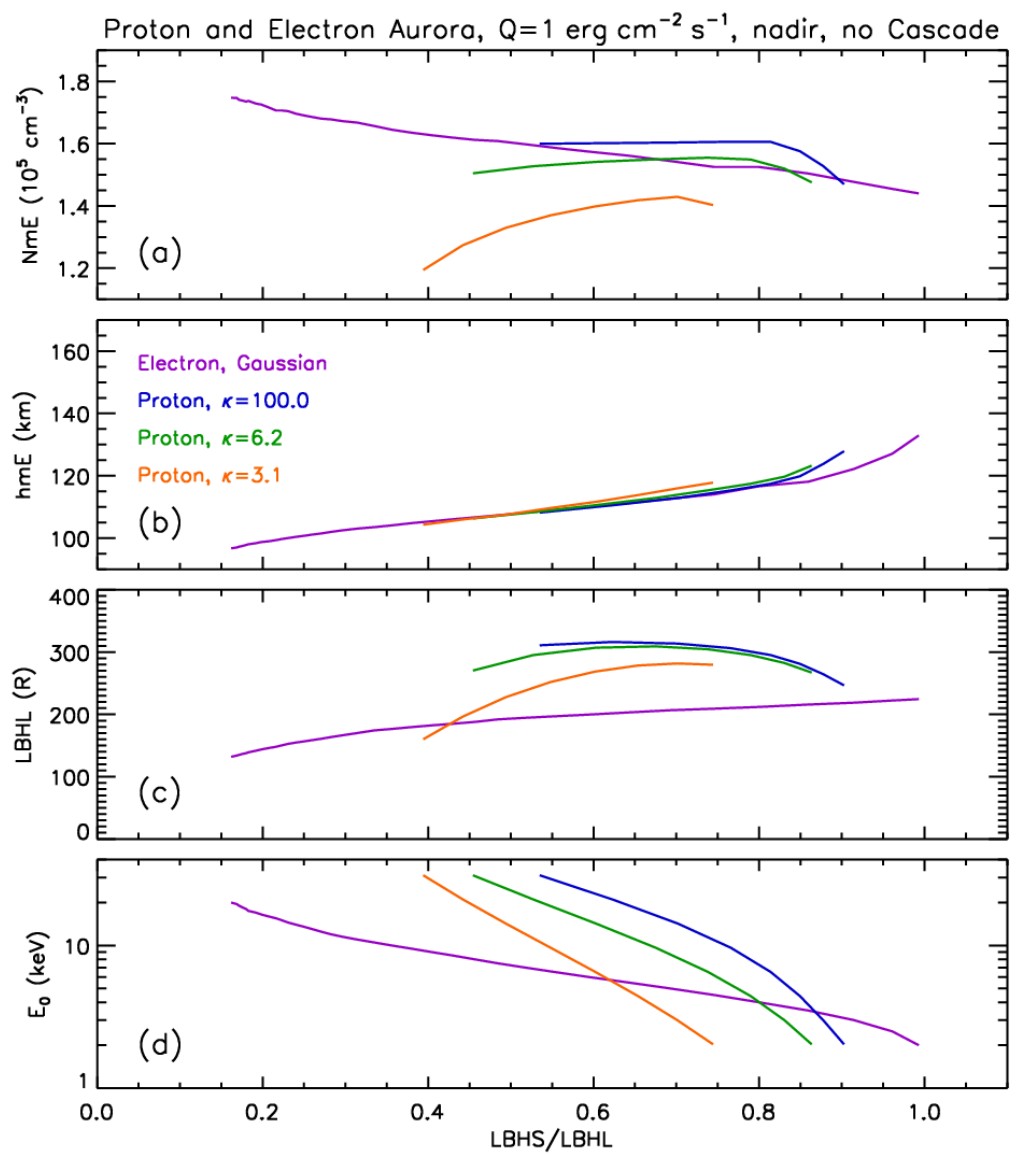

**Figure 1: Model-generated auroral NmE (a), hmE (b), LBHL (c), and $E_0$ (d) plotted versus LBHS/LBHL for pure electron aurora with a Gaussian spectral shape and pure proton aurora with a kappa function shape for three different kappa values (100, 6.2, and 3.1). Only values for nadir viewing are shown. LBH emissions do not include cascade contributions, as explained in the text.**

Mixed proton and electron aurora is actually more common than pure proton aurora (see, e.g., *Knight et al.* [2012] Fig. 4), but Fig. 1 at least gives an idea of what is expected from the two different types of aurora. LBH radiances resulting from simultaneous electron and proton precipitation are simply the linear combination of the radiances that would result from the two types of precipitation separately. Also, as discussed in Section 2.3 of *Knight et al.* [2018], under the assumption of local equilibrium between E region ionization and recombination processes, electron density is approximately proportional to the square root of the sum of squares of electron densities that would result from two different sources of ionization (i.e., electron and proton aurora, in this case). For further explanation, the reader is referred to *Zhang et al.* [2010] eqs. (1)-(5), which deals with the conceptually similar situation in which the two sources of ionization are electron precipitation and solar photo-ionization. In fact, since auroral ionization rates change linearly with changes in incident precipitating particle flux (e.g., *Basu et al.* [1987]), it follows in the same way from the assumption of local equilibrium that, for a given particle type and incident particle flux spectral shape, E region electron density at a given altitude will be approximately proportional to the square root of $Q$ (i.e., total incident energy flux) for that particle type.

Figure 1a shows that in near-Maxwellian cases, proton aurora produces NmE values that are similar to those produced by Gaussian electron spectra for the same LBHS/LBHL ratio, while proton auroral NmE decreases for wider proton spectra with the same energy flux. The hmE values as functions of LBHS/LBHL are similar for all four cases shown in Fig. 1b. For the two narrower proton spectra, LBHL values are ~50% (that is, a factor of ~1.5) higher than for electron spectra with the same precipitating energy flux and LBHS/LBHL values.

Taken in combination, these model results imply that the auroral FUV algorithm that assumes pure electron flux will give NmE values that are too high by a factor of $\sim\sqrt{1.5} = 1.22$ (taking the square root of 1.5, the typical ratio of LBHL emission efficiencies just stated) for pure proton flux spectra when the proton spectra are narrow, that is, $\kappa \geq \sim6.2$. The situation is more complicated for smaller $\kappa$ (e.g., $\kappa = \sim3.1$), since these differ with the narrower spectra in both NmE and LBHL, but it can be seen that, in general, there will still be an auroral FUV algorithm bias in the vicinity of 1.22 or greater for the wider type. To illustrate this, consider the case where $\kappa = 3.1$ and LBHS/LBHL=0.42, where the orange and purple lines cross in Fig. 1c. The proton and electron LBHL yields are the same in this case, but it can be seen in Fig. 1a that at LBHS/LBHL=0.42 the NmE value for the purple curve is ~1.3 times that of the orange curve, so the FUV-based remote sensing algorithm will overestimate NmE by a factor of ~1.3 (noting that 1.3 > 1.22) in this case if the emission is produced by proton aurora with $\kappa = 3.1$.

The above discussion gives a general idea of why the auroral FUV remote sensing algorithm for NmE is expected to be biased by a factor of at least 1.22 for pure proton aurora, but in order to predict the bias for ensembles of observations, it is

necessary to consider mixed proton and electron aurora, with varying spectral shapes. In Section 4.1, a statistical approach will be applied to ensembles of (nearly) coincident FUV and ionosonde observations. The results of Section 4.1 will be evaluated in the appendix in terms of a statistical simulation based on actual in situ ion (treated as proton) and electron flux observations, including realistic distributions of spectral shapes and electron and proton energy fluxes.

We include model-predicted hmE values in Fig. 1b for the sake of completeness, although comparisons of FUV-derived and ionosonde-observed hmE are not included in this work for the reasons mentioned in Section 1. It can be inferred from Fig. 1b that there should be little bias in derived hmE for pure proton aurora.

The proton auroral LBH yields on which Fig. 1 is based are the same as those described in *Knight et al.* [2008, 2012], and a comparison with electron and proton auroral emissions reported for other models (e.g., by *Hubert et al.* [2001]) was made in *Knight et al.* [2008] Section 5.2. The proton auroral hmE and NmE values for the blue curves (i.e., near Maxwellian) shown in Fig. 1 may be compared with electron density profiles shown in *Galand and Richmond* [2001] Fig. 1b. They give electron density for incident proton energy flux equal to 1 mW m$^{-2}$, which is the same as 1 erg cm$^{-2}$ s$^{-1}$. Their NmE values are all ~1.8x10$^5$ cm$^{-3}$ for Maxwellian proton flux spectra with E$_0$ values from 1 keV to 20 keV, and their hmE values go from ~130 km at E$_0$=1 keV to ~116 km at E$_0$=20 keV. Their NmE values are ~12.5% larger than our values of ~1.6x10$^5$ cm$^{-3}$ shown in Fig. 1a, and their hmE values are similar to our values shown in Fig. 1b. Some comparisons were made between in situ ion (treated as proton) flux data and E region densities predicted by particle transport models for proton aurora in *Basu et al.* [1987] Fig. 8, but the electron densities shown there seem puzzlingly low.

While it was already apparent from model results that proton precipitation would cause inaccuracies in auroral remote sensing methods based on LBH emissions (and assuming pure electron precipitation), it was implied by some studies comparing coincident auroral LBH observations and in situ electron and ion (treated as proton) flux data that the inaccuracies would be much worse than was predicted by models. These studies all involve SSUSI and Special Sensor J/5 (SSJ5) observations made from aboard the same satellites, as described in *Aerospace et al.* [2006,2011], *Knight et al.* [2008], and *Knight et al.* [2012]. (Also, *Correira et al.* [2011] revised some results from *Knight et al.* [2008].) These studies all concluded that B3C and other auroral transport and emission models were underestimating the efficiency of proton aurora (per unit energy flux) in producing LBH emission. The most likely explanation for the apparent underestimate was thought to be inaccurate LBH emission cross sections for proton and hydrogen impact, although other factors, which will be discussed shortly, could be involved.

SSJ5 is an in situ particle flux detector that observes precipitating electron and ion (treated as proton) fluxes in orbit at the location of the instrument. Additional details about the SSJ5 instrument may be found in *Knight et al.* [2008], *Emery et al.* [2008], and *Aerospace et al.* [2006,2011]. Since SSJ5 does not observe fluxes at energies above 30 keV, and since most of the auroral proton flux is often at energies above 30 keV, it was necessary to use proton extrapolation methods in the above

listed comparisons. Two types of extrapolation methods were involved: an earlier climatological method [*Aerospace et al.*, 2006; *Knight et al.*, 2008; *Correira et al.*, 2011] and a later non-climatological method [*Aerospace et al.*, 2011] and *Knight et al.* [2012]. Some of the model assumptions underlying the later non-climatological extrapolation method will be considered in Section 4.2.

In the comparisons just listed, monoenergetic emission feature yield curves derived from B3C model results (see *Knight et al.* [2012] Fig. 1) were used to derive radiances from SSJ5 flux observations by integrating the products of the monoenergetic yield curves with the SSJ5 fluxes. (See eq. (1) of *Knight et al.* [2012] and the surrounding text for additional information on this method.) The radiance values obtained from SSJ5 fluxes were then compared with coincident radiances obtained by averaging SSUSI L1B pixels over coincident areas. These values will be referred here to as $J5_e(LBHS)$, $J5_p(LBHS)$, and SSUSI(LBHS), and likewise for LBHL. Here, the e and p subscripts refer to electron and ion (treated as proton) flux. Extrapolation above 30 keV is included in the derivation of radiances for proton fluxes. The statistical SSUSI/SSJ5 ratios $f_e^{LBHS}$ and $f_p^{LBHS}$ are values fitted to the overdetermined equation, e.g.,

$$SSUSI(LBHS) \approx f_e^{LBHS}J5_e(LBHS) + f_p^{LBHS}J5_p(LBHS), \tag{1}$$

(and similarly for LBHL) for an ensemble of SSUSI-SSJ5 coincidences. See *Knight et al.* [2008, 2012] for a description of the fitting method. The SSUSI/SSJ5 statistical ratios describe the level of agreement between SSUSI, SSJ5 and the model (B3C in our case) in terms of two multiplicative factors, which represent the combined effect of model and instrument calibration errors (if any). In fact, the systematic errors may not be purely multiplicative, but it is reasonable to assume that they are as a first order approximation. Under ideal conditions, with exact models, perfect coincidences, perfect extrapolation, no noise, and no instrument calibration errors, one would expect that $f_e^{LBHS}$ and $f_p^{LBHS}$ would both be unity and that the left side of (1) would equal the right side for every coincident observation.

Define $f_{p/e}^{LBHS} = f_p^{LBHS}/f_e^{LBHS}$ and $f_{p/e}^{LBHL} = f_p^{LBHL}/f_e^{LBHL}$. These ratios of ratios are more useful than individual statistical SSUSI/SSJ5 ratios for describing model underestimates of proton LBH emission efficiency, since SSUSI calibration errors (if any) tend cancel out in the ratios. In the non-climatological extrapolation method, moreover, it is assumed that $f_{p/e}^{LBHS} = f_{p/e}^{LBHL}$. (See *Knight et al.* [2012] for the reasoning behind this assumption.) Under this assumption, let $f_{p/e}^{LBH}$ denote the value of the two ratios, which will be referred to as a proton/electron model LBH bias. Practically speaking, any estimate of proton/electron model LBH bias from actual observations is affected by in situ flux detector bias, proton extrapolation error, and other issues discussed in *Knight et al.* [2008, 2012], but it is still convenient for our purposes to be able to think of it as a "model bias" for which the estimation methods themselves, in practice, can be biased.

Proton/electron model LBH biases of 3.00 (*Knight et al.* [2012] Table 4), 2.31 (*Aerospace et al.* [2011] Fig. 7.1-14), and 1.84 (*Gelinas and Hecht* [2016]) were found in F16, F18, and F19 (respectively) SSUSI-SSJ5 comparisons. (The differences between these values for F16, F18, and F19 are accounted for to some extent by minor changes in the analysis method, which are explained in *Gelinas and Hecht* [2016]. These changes have to do with refinements in how the SSUSI instrument

response to spectral radiance is modelled.) For the sake of discussion, we will say that the proton/electron model LBH bias implied by these results in combination is $f_{p/e}^{LBH} = \sim 2$. Since the B3C model already predicts that proton aurora is a factor of $\sim 1.5$ more efficient in producing LBH than electron aurora (as described earlier in this section), $f_{p/e}^{LBH} = 2$ implies that proton aurora is actually $2 \times 1.5 = 3$ times as efficient as electron aurora in producing LBH aurora. Since the auroral FUV remote sensing algorithm (see *Knight et al.* [2018] eq. (2)) sets NmE in proportion to the square root of LBHL (which is justified by

the above observations on electron density being approximately proportional to the square root of $Q$), it is implied that the algorithm should give NmE values that are a factor of approximately $\sqrt{3} = 1.73$ too high for pure proton aurora.

The SSUSI-SSJ5 comparison method described above is designed in such a way as to allow SSUSI calibration errors (if any) to drop out, but it is still sensitive to SSJ5 calibration errors (if any). It is especially sensitive to SSJ5 ion flux calibration errors at the two highest energy channels, i.e., ~20 keV and 30 keV.

**4 FUV-Ionosonde Proton Analysis**

**4.1 Statistical results for NmE**

As described in Section 3, based on model results, it is expected that FUV-derived auroral NmE (i.e., derived under the assumption of pure electron aurora) will be too high for pure proton aurora by a factor of ~1.22, and based on SSUSI-SSJ5 comparisons, it is expected that the proton bias factor will have a much larger value of ~1.73. It turns out, however, that our

FUV-ionosonde comparisons indicate that there is no proton auroral bias in FUV-derived NmE.

Let the ionosonde-coincident radiances be denoted by the instrument name, followed by a subscript indexing the coincidence number, followed by the FUV channel in parentheses. For the 1216 channel, "bc" will be added to indicate that the radiances have been corrected for geocoronal background. For example, GUVI$_i$(1216,bc) is the i[th] coincident background-corrected GUVI 1216 value, and F16-SSUSI$_i$(LBHS) is the i[th] coincident F16-SSUSI LBHS value. When the whole set of coincident

values for one instrument is being represented, the index subscript will be omitted. The exact relative contribution of precipitating proton energy flux to the total incident proton and electron energy flux (or to any of the channel radiances) cannot be determined from the radiances, but a variable can be defined in terms of the radiances that at least tends to increase or decrease with the relative level of proton energy flux. This proton-indicating variable (PIV) will be denoted by GUVI$_i$(PIV), F16-SSUSI$_i$(PIV), and F18-SSUSI$_i$(PIV) and is given by the formula

$$270 \quad \text{GUVI}_i(\text{PIV}) = \frac{\frac{\text{GUVI}_i(1216,\text{bc})}{\text{median}(\text{GUVI}(1216,\text{bc}))}}{\frac{\text{GUVI}_i(1216,\text{bc})}{\text{median}(\text{GUVI}(1216,\text{bc}))} + \frac{\text{GUVI}_i(\text{LBHL})}{\text{median}(\text{GUVI}(\text{LBHL}))}} \quad (2)$$

for GUVI and likewise for the other two instruments. For nonnegative radiances, this will take on values between zero and one. The purpose of normalizing by the medians is to reduce the effects of possible calibration differences between the instruments so that the proton-indicating variable takes on the same approximate range of values for all three instruments. It also turns out that the normalization causes the PIVs to have a median value of approximately 0.5 for all three instruments, 275 which simply means that the two conditions

$$\frac{\text{GUVI}_i(\text{LBHL})}{\text{median}(\text{GUVI}(\text{LBHL}))} > \frac{\text{GUVI}_i(1216,\text{bc})}{\text{median}(\text{GUVI}(1216,\text{bc}))}, \quad (3)$$

$$\frac{\text{GUVI}_i(\text{LBHL})}{\text{median}(\text{GUVI}(\text{LBHL}))} < \frac{\text{GUVI}_i(1216,\text{bc})}{\text{median}(\text{GUVI}(1216,\text{bc}))}, \quad (4)$$

are approximately equally likely, and likewise for the other two instruments. There are a number of reasons why the PIV does not exactly determine the relative level of proton precipitation. As can be seen in Fig. 1 of *Knight et al.* [2012], the Ly-α 280 yield varies with precipitating particle energy. Also, electron auroral NI 120.0 nm emission contributes to the instrument response in the Ly-α channel, as discussed in *Knight et al.* [2008, 2012]. Regardless, it is clear that the PIV must be statistically associated with proton aurora.

If there were a bias in FUV-derived NmE associated with proton aurora, then it would be expected that the PIV would provide some extra information, so that in combination with FUV-derived NmE it would allow a better prediction of 285 ionosonde-observed NmE. In order to determine whether this is the case, we first use regression to obtain an estimate of ionosonde-observed NmE in terms of FUV-derived NmE and then study the statistical relationship between the PIVs and the residuals.

We now introduce some additional notation. The FUV-derived NmE values are denoted by, e.g., $\text{GUVI}_i(\text{NmE,F})$, where "F" stands for "FUV", and the coincident ionosonde-observed values are denoted by, e.g., $\text{GUVI}_i(\text{NmE,I,pre})$, 290 $\text{GUVI}_i(\text{NmE,I,post})$, where "I" stands for "ionosonde" and "pre" and "post" indicate the ionosonde observations immediately before and after (respectively) the time of coincidence. (It may seem counter-intuitive to label ionosonde observations according to coincident FUV instruments, but this approach makes sense here because our transformations of data values are specific to FUV instruments and not ground stations.)

Our regression method is as follows. To reduce the effect of outliers, we convert to log space and use least absolute deviation 295 regression to fit coefficients, e.g., $\text{GUVI}(\alpha, \text{pre})$, $\text{GUVI}(\beta, \text{pre})$, to the data as follows

$$\ln\big(\mathrm{GUVI}_i(\mathrm{NmE, I, pre})\big) \approx \mathrm{GUVI}(\alpha, \mathrm{pre}) + \mathrm{GUVI}(\beta, \mathrm{pre}) \ln\big(\mathrm{GUVI}_i(\mathrm{NmE, F})\big), \tag{5}$$

and likewise for post-coincidence ionograms and the other two FUV instruments. These fitted lines are illustrated by Fig. 2, which shows ionosonde-observed NmE (y axis) plotted versus FUV-derived NmE values (x axis) with the fitted lines. This figure is similar to Fig. 6 and Fig. 7 from *Knight et al.* [2018], with these differences: 1. there are fewer F16 points because a 300 R threshold is used for F16; 2. the x and y axes have been reversed; 3. this figure shows fitted lines; 4. a few GUVI and F18 points are excluded here because of extra conditions described in Section 2.1.

The reader is referred to *Knight et al.* [2018] for a discussion of the consistency between the FUV and ionosonde observations. The exact values of the parameters of the fitted lines in Fig. 2 are not of direct interest here. The purpose here is to determine whether the PIV adds extra information (for predicting ionosonde-derived NmE) that is not already provided by FUV-derived NmE. At a basic, conceptual level, the approach to be used here is similar to the approach of comparing reduced and full models described in *Rencher and Schaalje* [2008, sections 7.10 and 8.2]. Our approach, to be described next, relies on a statistical simulation to establish a result rather than a formal mathematical method, such as the *F* test of *Rencher and Schaalje* [2008, section 8.2].

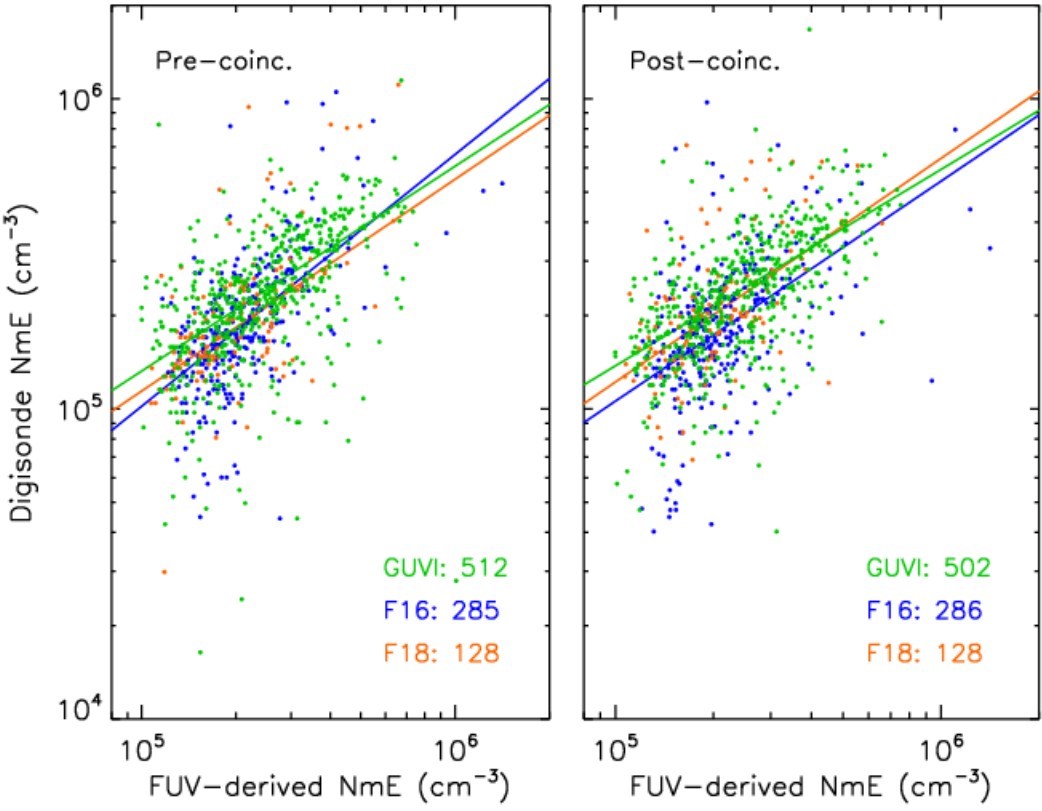

**Figure 2: Ionosonde-observed NmE (y axis) plotted versus FUV-derived NmE for pre-coincidence (left) and post-coincidence Digisonde observations. Color-coded lines for the three satellites have been fitted in log space to the x and y values using least absolute deviations, as described in the text, for the purpose of determining residuals, as described in the text.**

We use the term "residual" to refer to the ratio of ionosonde-observed NmE to the value estimated from FUV-derived NmE by applying the exponential function to the right side of eq. (5). The notation, e.g., $GUVI_i(Res(NmE), pre)$ will denote the

315 residual value given by

$$GUVI_i(Res(NmE), pre) = \frac{GUVI_i(NmE, I, pre)}{exp\big(GUVI(\alpha, pre) + GUVI(\beta, pre) \ln\big(GUVI_i(NmE, F)\big)\big)}. \tag{6}$$

This generalized type of residual, which will sometimes be referred to simply as $Res(NmE)$, is dimensionless, since the right side of eq. (6) has a ratio of NmE values. Although "residuals" usually refer to differences and not ratios, note that a ratio can be interpreted as a difference in logarithmic space.

When data values for all three FUV instruments are combined, they will be referred to with the same notation as before, except with the instrument abbreviation replaced by "COMB." The calculations of residuals and PIVs are still instrument-specific and are performed prior to combining the derived values. Let $COMB(Res(NmE), PIV < 0.5, pre)$ and $COMB(Res(NmE), PIV \geq 0.5, pre)$ be the subsets of $COMB(Res(NmE), pre)$ for which PIV is or less than 0.5 and greater than or equal to 0.5, respectively. These will be referred to as "below" and "above" sets. Similar definitions can be made for

the post-coincidence ionograms and the individual FUV instruments. The statistical effect of proton aurora on $Res(NmE)$ will now be described in terms of the ratio of medians

$$COMB(BAR, pre) = \frac{median\big(COMB(Res(NmE), PIV < 0.5, pre)\big)}{median\big(COMB(Res(NmE), PIV \geq 0.5, pre)\big)}, \tag{7}$$

where BAR stands for below-to-above ratio, and similarly for post-coincidence ionograms. The BAR quantity will be referred to as a "statistic" because it is an aggregate property of the sets of values being compared.

Figure 3 shows PIV (x-axis) and $Res(NmE)$ (y-axis) values derived from FUV-ionosonde coincidences for pre-coincidence (upper panel) and post-coincidence (lower panel) ionograms. The numbers of points included in $COMB(Res(NmE), PIV < 0.5, pre)$, etc., are printed in the lower left and right corners of both panels. The geocorona removal method (see Section 2.1) results in some $COMB(1216, bc)$ values that are less than zero, which is why negative PIV values occur. PIV values below $-0.25$ are set to $-0.25$. The below and above medians in the numerator and denominator (respectively) of eq. (7) are plotted

as gray horizontal solid lines, and it can be seen they are all close to unity. In fact, all four medians are within 0.015 of unity, and the BARs are $COMB(BAR, pre) = 1.012$ and $COMB(BAR, post) = 0.974$. Since the two BAR statistics are on opposite sides of unity, it is immediately evident that there is no indication of a bias associated with proton aurora. BARs for the individual instruments (not shown) were also computed, and they are near unity as well.

The significance of the medians of the four sets, i.e., $\text{COMB}(\text{Res}(\text{NmE}), \text{PIV} < 0.5, \text{pre})$, $\text{COMB}(\text{Res}(\text{NmE}), \text{PIV} \geq 0.5, \text{pre})$, $\text{COMB}(\text{Res}(\text{NmE}), \text{PIV} < 0.5, \text{post})$, and $\text{COMB}(\text{Res}(\text{NmE}), \text{PIV} \geq 0.5, \text{post})$, can be understood as follows. *Van der Vaart* [1998] (pgs. 54-55) states that the sample median of a distribution is asymptotically normal (letting the sample size $n$ tend to infinity) with an asymptotic variance (i.e., the limit as $n \to \infty$ of the variance multiplied by $n$) equal to $\left(2f(\theta_0)\right)^{-2}$, where $f$ is the probability density function of the underlying distribution and $\theta_0$ is the median of the distribution. Now we convert to log space by taking the natural logarithms of all of the Res(NmE) values. We let the null hypothesis be that any of the four sets is sampled from distributions with medians equal to unity, which implies medians equal to zero in log space, i.e., $\theta_0 = 0$. The distribution standard deviations, denoted by $\sigma_d$, of $\ln\left(\text{Res}(\text{NmE})\right)$ in all four sets are estimated from the data as ~0.4. We use a normal approximation and plug $\sigma_d$ into the probability density function for a normal distribution to obtain $f(\theta_0) \approx (2\sigma_d^2\pi)^{-1/2} = {\sim}1.00$. (It is just a coincidence that $f(\theta_0)$ is close to unity.) It is implied that the sample median has a standard deviation, denoted by $\sigma_m$, approximately equal to $\left(2f(\theta_0)\right)^{-1} n^{-1/2}$. Since $n \approx 450$, this gives $\sigma_m \approx$ 0.024. Since the medians are within 0.015 of unity, their natural logarithms are within ~0.015 of zero. Since the natural logarithms of all four medians are all well within $\pm\sigma_m$, the null hypothesis is accepted for any confidence interval including $\pm\sigma_m$ in a normal distribution with standard deviation equal to $\sigma_m$. This means that none of the four medians is significantly far from unity and that therefore there is no bias associated with proton aurora.

Given biases mentioned at the beginning of this section of ~1.22 (predicted by the B3C model) or ~1.73 (predicted by earlier in situ comparison results) expected for auroral FUV-derived NmE in the presence of proton precipitation, *one would expect a BAR statistic greater than unity*. This is because FUV-derived NmE is expected to overestimate actual NmE when a significant portion of the observed LBH is produced by proton precipitation, meaning that $\text{COMB}(\text{Res}(\text{NmE}), \text{pre})$ and $\text{COMB}(\text{Res}(\text{NmE}), \text{post})$ should tend to be lower when there is more proton precipitation, as indicated by the condition $\text{PIV} > 0.5$. This leads to the following question: What is the likelihood that BAR statistics so close to unity would be obtained if the predicted FUV-derived NmE biases of ~1.22 or ~1.73 for proton aurora were correct? In the appendix, an answer will be obtained for this question using a statistical simulation incorporating synthetically generated sets of FUV and ionosonde observations. It will be shown there that the probability of obtaining a BAR statistic of 1.012 or 0.974 (see above) is essentially zero, given either of the predicted NmE biases just mentioned. Given the B3C results described in Section 3, the simulation described in the appendix predicts a BAR statistic of $1.121 \pm 0.030$, and given the in situ comparison results, also described in Section 3, calling for proton auroral LBH emission yields to be increased by a factor of ~2, the simulation described in the appendix predicts a BAR statistic of $1.271 \pm 0.036$.

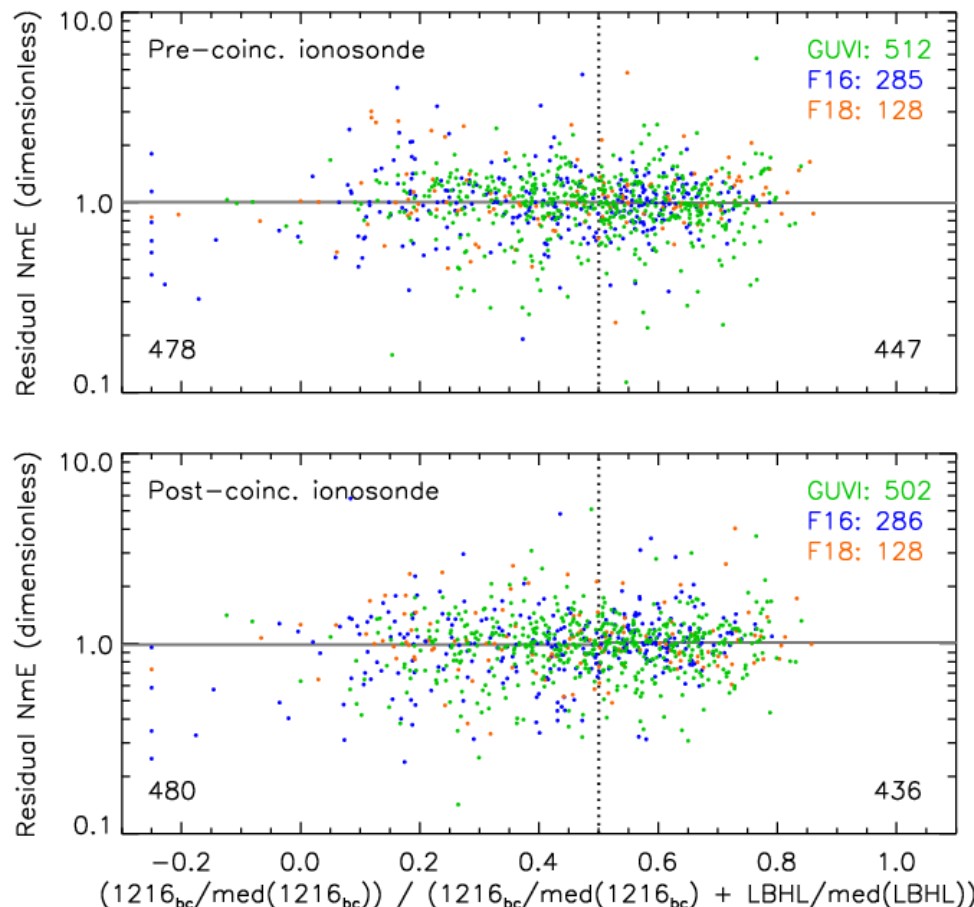

**Figure 3: Residual NmE (y-axis, as described in the text) plotted versus the ratio of background-corrected 1216 to the sum of background-corrected 1216 and LBHL, where the radiances are normalized by their medians. The upper and lower panels show results for ionosonde observations immediately before and after (respectively) the times of satellite coincidence with the ground stations. Horizontal gray lines indicate the median residual values in left and right bins on either side of the dotted vertical line at x=0.5. The numbers of points in the bins are displayed in the lower left and right corners of the panels. The numbers of points shown for each satellite are given in the upper right corners of the two panels.**

### 4.2 Possible explanations for unexpected NmE result

As just described in Section 4.1, the following BAR statistics, as defined by (7), were found for the FUV-ionosonde coincidences: $COMB(BAR, pre) = 1.012$ and $COMB(BAR, post) = 0.974$. The proximity of these values to unity means that there is essentially no statistical difference in the accuracy of the auroral FUV algorithm giving NmE between the two complementary sets with $PIV < 0.5$ and $PIV \geq 0.5$, where PIV is the proton-indicating variable defined by (2). The PIV is defined in such a way as to be highly correlated with the ratio of precipitating proton energy flux to total electron and proton energy flux. This is because HI 121.6 nm emission is produced by proton aurora but not electron aurora. BAR values near unity mean that there is no apparent bias in the auroral FUV remote sensing algorithm giving NmE (assuming electron aurora) associated with proton precipitation, which is surprising given that auroral particle transport and emission models

give LBH yields for proton aurora that are ~50% higher than those for electron aurora. The simulation results described in
the appendix suggest that BAR statistics in the vicinity of 1.12 should have been obtained if the B3C model results were correct and that somewhat larger BAR statistics in the vicinity of 1.27 should have been obtained if the in situ comparison results (e.g., *Knight et al.* [2012]) calling for an increase in proton auroral LBH yields by a factor of ~2 were correct.

Broadly speaking, there are two approaches by which one can attempt to explain the unexpected finding. The first is to posit that the earlier in situ comparison results are incorrect. The second is to look for flaws in the methodology of the current
study, involving differences between the two types of observations (i.e., FUV and ionosonde). For the moment, the first approach will be considered. Within this approach one can either attempt to attribute the discrepancy to potential model errors or attempt to identify potential flaws in the in situ comparison methodology or in the in situ data themselves. Setting aside the in situ comparison results for the moment, we infer from B3C model results and a statistical simulation, just mentioned above, that BAR statistics in the vicinity of 1.12 should have been obtained, as opposed to the values near unity
which were actually obtained. This discrepancy could be accounted for by one or both of the following two types of inaccuracies in the B3C model: either an overestimate of proton auroral LBH or an underestimate of E region ionization produced by proton aurora. These possibilities will not be considered further here, but it should be noted that similarities between proton auroral LBH emission and E region ionization predicted by B3C and other auroral particle transport and emission models were described in Section 3.

Now we continue with the first approach to understanding the unexpected finding by attempting to identify flaws underlying the earlier in situ results. As described earlier, the in situ comparison results relied on extrapolation of ion (treated as proton) flux above 30 keV, using a non-climatological method described in *Knight et al.* [2012]. As already mentioned, the extrapolation method is sensitive to SSJ5 ion (treated as proton) calibration errors in the two highest energy channels. As an example, if SSJ5 ion fluxes were 10% low at 20 keV and 30 keV, this would imply an extrapolation underestimate by
approximately 10%.

The extrapolation method is constrained by model assumptions, again as described in *Knight et al.* [2012]. One type of assumption is the selection of the model atmosphere, which affects both the modelled emission yields and resulting atmospheric ionization (although it is the emission yields that are relevant to the extrapolation method). The model atmosphere used in the model runs on which the extrapolation method is based is the same as the one described in Section
2.3 of *Knight et al.* [2018]. It is specified by a number of parameters, including the day of the year, local time, latitude, the geomagnetic Ap index, and the F10 index for solar activity. The extrapolation method gives slightly different answers depending on which parameter values are selected, but these changes cannot account for the discrepancy between the FUV-ionosonde and FUV-in situ comparison results. For example, in Appendix C2 of *Knight et al.* [2012] it is described how different values of F10 (i.e., 80 and 150) can only account for a ~10% change in extrapolated proton flux. The effects of
other possible model assumptions errors have been considered in detail in *Knight et al.* [2008,2012], *Correira et al.* [2011],

and *Aerospace et al.* [2011], and no likely explanation for the discrepancy with the current results stands out. Moreover, we have filtered the FUV-ionosonde comparison sets by a number of different criteria (e.g., day of year, local time, latitude, Kp, phase of solar cycle) in the hopes that this would point to some particular type of assumption error, but it did not.

One assumption involved in the proton flux extrapolation method (for the earlier in situ comparisons) that is worth reviewing
in particular, however, is the assumption that the ion species responsible for ion fluxes observed by SSJ5 are only protons, as opposed to other types of ions, like O$^+$. SSJ5 does not distinguish between different ion species. One way to test for the effect of the ion species assumption is to look for a change in the result when the comparison sets are filtered by the disturbance storm time (Dst) index as discussed in paragraph [64] of *Knight et al.* [2008]. We tried filtering by Dst in earlier in situ comparisons (i.e., *Knight et al.*, 2008) and found that it did not affect the results. This particular test was not done for
the current work, although we did filter by Kp, as mentioned above, and did not find an effect. Regardless, the possibility that non-proton ion precipitation is responsible for the unexpected results, either in the earlier in situ comparisons or in the current FUV-ionosonde comparisons, merits further investigation.

Next, the other approach to explaining the unexpected finding, i.e., to look for flaws in the methodology of the current study, will be considered. There are a number of types of differences between the FUV and ionosonde observations, and both
spatial and temporal non-uniformity exists in the aurora. The coincident observations are made at times offset by up to 15 minutes from each other. As described in *Knight et al.* [2018] Section 2.2, both types of observations gather information from a 30 km radius area over each ground station, but there are clearly differences in how each type of observation samples from that area. Nonetheless, a strong statistical association was found in *Knight et al.* [2018] between FUV-derived and ionosonde-observed NmE. Any attempt to dismiss the lack of a proton effect described in Section 4.1 as being caused by
auroral non-uniformity and differences between the two types of observations must somehow account for why a strong statistical association between the two types of observations was found in *Knight et al.* [2018].

One might suppose that the difference between the BAR statistics near unity obtained in Section 4.1 and the expected value of 1.12 (see the appendix) are not statistically significant, but the simulation in the appendix was designed in such a way as to show how the BAR statistic varies as a result of randomness associated with counting statistics and auroral variability in
time, and a BAR of ~1.00 was shown to be well outside the range of likely values.

We expected that FUV-derived NmE would be biased high relative to ionosonde-observed NmE in the presence of proton aurora, but this effect was not seen. In order for the expected bias to be hidden, there would have to be some difference between electron and proton aurora that affects the FUV and/or ionosonde observations in such a way as to cancel out the expected bias. We have not thought of any plausible explanation involving this type of effect, but some ideas along these
lines will now be discussed.

One difference between electron and proton aurora is that proton/hydrogen horizontal beam spreading (e.g., *Fang et al.*, 2007) occurs for the latter. While precipitating electrons follow the magnetic field lines to the emitting region, precipitating protons/H-atoms also move in directions perpendicular to the field lines, to some extent. In the B3C model runs done for this analysis, it was assumed that there was no horizontal spreading, but there is no obvious way in which this could have led to the unexpected finding in the FUV-ionosonde comparisons. In the actual physical setting, horizontal spreading would affect both emission and ion production in similar ways, so one would not necessarily expect a bias between LBH emission and ion production to result.

Another difference between electron and proton aurora is that electron aurora can be either discrete or diffuse, while proton aurora can only be diffuse [*Newell et al.*, 2009]. Discrete aurora tends to be more variable, and one might hypothesize that the difference between discrete and diffuse aurora, affecting the two observation methods in different ways, is somehow hiding the expected proton biasing effect. As is well known, discrete and diffuse aurora tend to occur towards the poleward and equatorward boundaries of the auroral oval, respectively. The magnetic latitudes of the ground stations (see *Knight et al.* [2018] Table 3) range from 61.1 to 66.8. Filtering by magnetic latitude did not affect the BAR statistics, which casts doubt on the notion that discrete aurora is responsible for the unexpected result. A number of other tests were tried, such as filtering by magnetic local time and other geophysical parameters.

## 5 Discussion and conclusion

A large number of coincident auroral FUV and ground-based ionosonde observations were compared to determine the statistical effect of proton precipitation on the accuracy of auroral FUV algorithms that assume pure electron precipitation. Examples of B3C model-generated proton and electron auroral radiances and E region parameters (i.e., NmE and hmE) were given in Section 3, and earlier FUV versus in situ precipitating particle flux comparison results, in which radiances are derived from particle fluxes, and which rely on proton extrapolation above 30 keV, were summarized to explain why a bias in auroral FUV-derived NmE was expected in the presence of proton precipitation.

Surprisingly, no proton-associated bias in FUV-derived NmE was found (Section 4.1). A statistical simulation using synthetic data, described in the appendix, indicates that this result would be quite unlikely given the differences in proton and electron LBH yields described in Section 3, particularly if the large proton-associated bias predicted by in situ comparison results (*Knight et al.* [2012]) were correct. In order to explain the lack of any bias for proton aurora, it would be necessary for either model-predicted proton auroral LBH yields to decrease or for model-predicted proton auroral NmE values to increase.

In light of this result, we cannot explain the large proton LBH bias predicted by in situ comparisons (e.g., *Knight et al.* [2012]). The proton flux extrapolation method used in *Knight et al.* [2012] is constrained by observations and assumptions

that cannot readily be modified. Possible explanations for the discrepancy were discussed in section 4.2. The proton flux extrapolation method used in *Knight et al.* [2012] is sensitive to calibration errors in the two highest SSJ5 ion energy channels. The possible role of model assumptions involved in the extrapolation method was considered. The possible effect of non-proton ion precipitation on the in situ results merits a closer look. The combined role of auroral variability and differences between the two types of observations was also considered, but no explanation was found there. New comparisons of LBH and in situ auroral particle flux data, using proton flux observations that do not require extrapolation above 30 keV (e.g., POES TED and MEPED), are called for.

The result that proton aurora does not introduce a bias in auroral FUV-derived NmE puts previous work that relied on auroral FUV-derived NmE on a firmer footing. This includes *Zhang et al.* [2010], in which auroral FUV-derived NmE is assimilated into an ionospheric model. This also applies to some extent to previous work in which auroral FUV-derived Hall and Pedersen conductances have been studied, including *Aksnes et al.* [2002] and *Coumans et al.* [2004], with the caveat that accuracy of NmE does not directly imply accuracy of conductances. Given the present result on NmE, it appears likely that auroral FUV-derived energy flux (i.e., Q) is not severely biased by proton aurora to the extent predicted by *Knight et al.* [2008, 2012]. Examples of previous work involving auroral FUV-derived Q include *Brittnacher et al.* [1997], *Liou et al.* [1998], *Newell et al.* [2001], *Hubert et al.* [2002], *Baker et al.* [2004], *Zhang and Paxton* [2008], and *Luan et al.* [2010].

### Author contribution

H. Knight carried out the analysis described herein.

### Competing interests

The author declares that he has no conflict of interest.

### Code availability

The routines for visualization, data analysis, and statistical simulation are available from the author upon request.

### Data availability

Ground station ionosonde data were provide by the Global Ionosphere Radio Observatory (GIRO) and are available at http://giro.uml.edu/. FUV L1B data may be obtained from http://guvitimed.jhuapl.edu/ and http://ssusi.jhuapl.edu/. All of the coincident FUV/ionosonde data values used in this work are provided at the website http://www.cpi.com/projects/fuvi.html.

**Acknowledgements**

This work was funded by a NASA grant, NNX13AF41G. The author acknowledges the operators of the Digisondes at Goose Bay (operated by the US Air Force), Gakona (operated by the US Air Force), and Tromso (operated and funded in part by QinetiQ). The study used the Norilsk ionosonde data of Center for Common Use «Angara» [http://ckp-rf.ru/ckp/3056] obtained with budgetary funding of Basic Research program II.12. The author acknowledges GIRO (http://spase.info/SMWG/Observatory/GIRO).


The author thanks Doug Strickland for reading the manuscript and making some helpful suggestions.

**Appendix A: Simulation to quantitatively evaluate hypotheses**

The FUV-ionosonde comparison results given in the Section 4.1 show no evidence of any bias (high or low) in LBH-derived NmE associated with proton aurora, which is surprising given that NmE bias factors of ~1.22 and ~1.73 are predicted for pure proton aurora based on model and in situ comparison results, respectively. In this appendix, a statistical simulation will be employed to determine the distribution of BAR statistics (see Section 4.1) that would be expected given conditions similar to the ones described for the actual FUV-ionosonde comparison sets.

Our statistical simulation uses synthetic FUV-ionosonde comparisons generated from National Oceanic and Atmospheric Administration (NOAA) Polar-orbiting Operational Environment Satellites (POES) 17 Total Energy Detector (TED) and Medium Energy Proton Electron Detector (MEPED) in situ electron and ion (treated here as proton) fluxes from 2003 and 2004 (*Evans and Greer* [2004]). LBHS, LBHL, and NmE values were derived for all of the POES observations using the B3C model with the exact same set of model assumptions that were used to generate Fig. 1. In this section, the statistical

ratio notation defined in Section 3 will be reused with a slightly different meaning. Here, it will refer to scaling factors applied to FUV values derived from particle fluxes using the B3C model. In this section, $f_e^{LBHS} = f_e^{LBHL} = 1$ and $f_p^{LBHS} = f_p^{LBHL} \equiv f_p^{LBH}$ (defining a new term, $f_p^{LBH}$). A particular value of $f_p^{LBH}$ is set at the outset of each simulation run.

The idea of the simulation is to generate synthetic sets of coincident FUV and ionosonde observations comparable with actual coincident sets described in section 4.1. Realism was sought in the simulation. Unfortunately, it would be too

complicated to describe exactly how the needed realism was achieved, and some details are omitted so as to avoid overburdening the reader.

There are three FUV sensors and four ground stations. Let "FS" and "GS" stand for any particular FUV sensor and ground station, respectively. (In other words, FS could be either GUVI, F16, or F18, and GS could be either at Goose Bay, Gakona,

Norilsk, or Tromso.) Let N(FS,GS) denote the number of coincident observations meeting the conditions described in Section 2 for FUV sensor FS and ground station GS. In each run of the simulation, for each FS and GS, N(FS,GS) POES observations are randomly selected subject to the conditions that $NmE > 2 \times 10^5$ cm$^{-3}$ and that the absolute value of the magnetic latitude of the POES satellite is within 0.1 degree of the magnetic latitude of the GS. A ground station NmE and a set of NI 120.0 nm, HI 121.6 nm, LBHS, LBHL values are obtained from values already computed for each POES observation by multiplying by lognormal random variates generated with a $\sigma$ width parameter selected so as to be consistent with the value StdDev(ln(y/x))=0.44 shown in *Knight et al.* (2018) Table 4 column A.II. This is to represent the random changes that take place in actual NmE going from the time of the ground observation to the time of the satellite observation, or vice versa. (Some details are omitted here, but the author will provide the missing details on request.) The proton auroral contributions to the simulated LBHS and LBHL values are multiplied by $f_p^{LBH}$. Additionally, the simulated emission values are modified by sampling from Poisson noise distributions to represent the random effects of counting statistics, using appropriate responsivities (see Section 2.1) and accounting for geocorona in the case of HI 121.6 nm.

Once the ground station NmE and satellite-observed emission values have been generated as just described, FUV-derived NmE is computed from the simulated LBHS and LBHL values using the two-channel auroral FUV remote sensing algorithm described in *Knight et al.* (2018) Section 2.3. Finally, the BAR statistic is computed in exactly the same way as described in section 4.1. One thousand of the above-described runs were done for $f_p^{LBH} = 1.0$ and $f_p^{LBH} = 2.0$ (each), with the BAR values saved for analysis. Normalized histograms showing the distributions of BAR statistics obtained this way are shown in Fig. A1.

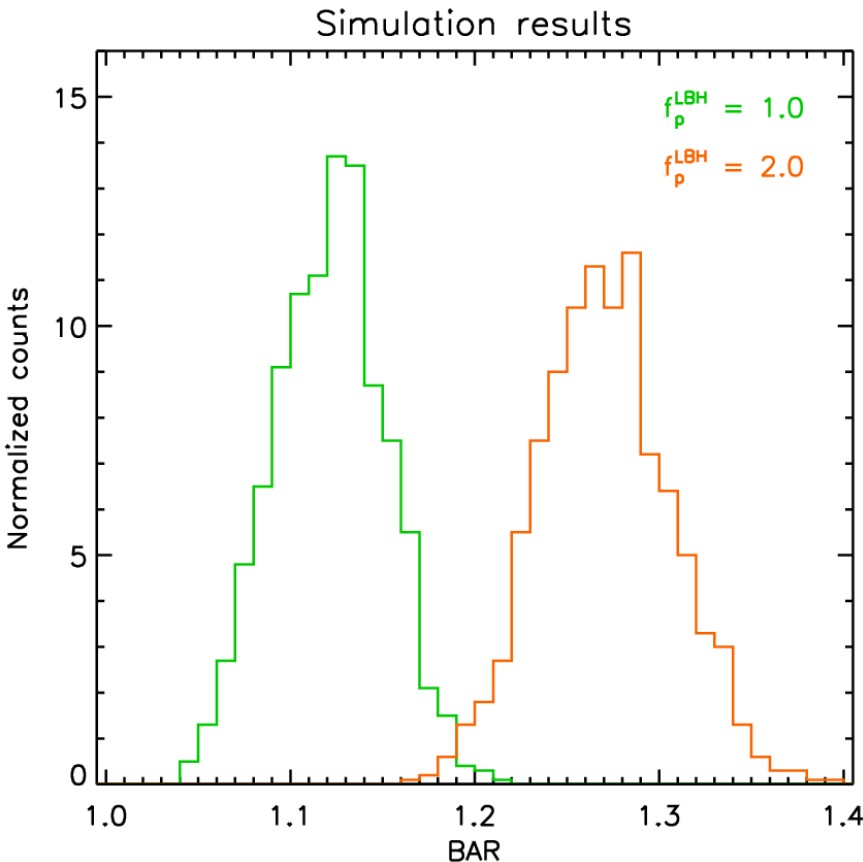

**Figure A1: Histograms of below-to-above ratios (BAR, defined in Section 4.1) from 1000 randomized simulation runs (each) for $f_p^{LBH} = 1.0$ (green) and $f_p^{LBH} = 2.0$ (orange). The counts (i.e., numbers of BAR statistics that fall in each histogram bin) are normalized so that the integrated area under each curve equals unity.**

The means of the BAR statistics for $f_p^{LBH} = 1.0$ and $2.0$ are $1.121$, and $1.271$, respectively, and the standard deviations are $0.030$ and $0.036$, respectively. The results for $f_p^{LBH} = 1.0$ correspond to the hypothesis that the B3C-predicted emission yields and NmE values for electron and proton aurora are correct, and the results for $f_p^{LBH} = 2.0$ correspond to the hypothesis that the B3C proton auroral LBH emission yields need to be increased by a factor of ~2.0, as suggested by earlier in situ comparison results described in Section 3.

Evidently, the BAR statistics of $1.012$ and $0.974$ for the pre-coincidence and post-coincidence comparisons described in Section 4.1 are below all of the BAR statistics obtained for the 1000 simulation runs generated for $f_p^{LBH} = 1.0$ and even farther below the BAR statistics obtained for $f_p^{LBH} = 2.0$. Possible reasons for this surprising result related to B3C model assumptions, the possible sources of error in the earlier in situ comparisons, and the nature of FUV-ionosonde comparisons, are discussed in Section 4.2.

Considering possible explanations more specific to the simulation, it could be suggested that higher BAR statistics are found in the simulation than in the actual FUV-ionosonde comparisons because the precipitating electron and proton fluxes observed by POES-17 in 2003 and 2004 are not consistent with those occurring at the actual overpasses of the ground stations by the three satellites. It can be seen in Table 2 of *Knight et al.* [2018] that about two-thirds of the overflights included in our study occurred from 2002 to 2007, during the descending phase of solar cycle 23, so the prevailing conditions should be similar. There was insufficient time to generate simulations based on observations made by other POES satellites in other years.

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
