# Peer review of "Auroral ionospheric E region parameters obtained from satellitebased far ultraviolet and ground-based ionosonde observations: Effects of proton precipitation"

_Annales Geophysicae, 2019_

## Referee Comment (RC1) · Anonymous Referee #1 · 15 Nov 2019

In this paper, the E-region electron density estimated from auroral fa ultraviolet (FUV) is compared with that observed with ionosondes. The obtained results could contribute to studies of aurora and ionosphere. Therefore, this paper is worth publishing in this journal. However, the following comments need to be addressed before its publication.

–ll. 141-142: Does "local equilibrium" means the equilibrium between the ionization and recombination rates of plasma in the E region? If so, it is better to describe so.

–ll. 143-144, "electron density is approximately proportional to the square root of the

sum of squares of electron densities that would result from two different sources of ionization (i.e., electron and proton aurora, in this case)":

This explanation seems strange. If production rates of the ion by electron and proton aurora ($q_e$ and $q_{pi}$, respectively) is balanced with the recombination rate ($a * n^2$, where a is a coefficient and n is the electron density), we can describe as $q_e + q_p = a * n^2$ Then, $n = 1/a * sqrt( q_e + q_p )$ Does the author consider that $q_e$ and $q_p$ are proportional to square of electron density?

---

## Author Comment (AC1) · 16 Nov 2019

Reviewer comment:

In this paper, the E-region electron density estimated from auroral fa ultraviolet (FUV) is compared with that observed with ionosondes. The obtained results could contribute to studies of aurora and ionosphere. Therefore, this paper is worth publishing in this journal. However, the following comments need to be addressed before its publication.

–ll. 141-142: Does "local equilibrium" means the equilibrium between the ionization

and recombination rates of plasma in the E region? If so, it is better to describe so.

HK reply:

Yes. I will add that to the text. I was leaving out details that were already given in Knight et al. (2018), and line 142 of the manuscript already refers to section 2.3 of Knight et al. 2018, in which more details about the nature of the equilibrium are given. Regardless, it is no problem for me to add a description of the equilibrium to the current paper under review.

Reviewer comment:

–ll. 143-144, "electron density is approximately proportional to the square root of the sum of squares of electron densities that would result from two different sources of ionization (i.e., electron and proton aurora, in this case)":

This explanation seems strange. If production rates of the ion by electron and proton aurora ($q_e$ and $q_{pi}$, respectively) is balanced with the recombination rate ($a * n^2$, where a is a coefficient and n is the electron density), we can describe as $q_e + q_p = a * n^2$ Then, $n = 1/a * sqrt( q_e + q_p )$ Does the author consider that $q_e$ and $q_p$ are proportional to square of electron density?

HK reply:

First, there is a minor notational issue, which is that I would not use "q" to denote an ion production rate, since the letter/variable q is reserved for precipitating particle energy flux (for example, see Knight et al., 2018). Aside from that, an equivalent equation to the ones you have written is given by Zhang et al. (2010), eqs. (2). The more complete citation, which is given in Knight et al. (2018), is:

Zhang, Y., Paxton, L. J., Bilitza, D., & Doe, R. (2010). Near real-time assimilation in IRI of auroral peak E-region density and equatorward boundary. Advances in Space Research, 46, 1055–1063. https://doi.org/10.1016/j.asr.2010.06.029

The only difference is that Zhang et al. are talking about the combination of ion production rates resulting from electron precipitation and solar EUV flux, while I am talking about the combination of electron and proton precipitation. The same principles apply in either case.

The statement at lines 143-144 you mentioned is equivalent to Zhang et al. (2010) eq. (5). I will add a reference to Zhang et al. eq. (5) to the manuscript in order to support the statement about electron density being approximately proportional to the square root of the sum of squares, etc.

As far as whether I think that individual production rates are proportional to the square of electron density, I would say yes, assuming that I am not misunderstanding your meaning. This is exactly what is stated in Zhang et al. eqs. (3) and (4).

---

## Referee Comment (RC2) · Anonymous Referee #1 · 19 Nov 2019

The author considers that individual production rates are proportional to the square of electron density. It should be reconsidered.

According to Zhang et al. (2010), the electron density $N_e$ is determined by chemical equilibrium between production and loss of ion. $P_{euv}$ and $P_e$ are the electron production rates due to solar EUV flux and electron precipitation, respectively. Due to the dissociative recombination, molecular ion, such as O₂+ and NO+ are recombined. This recombination process can be written as

$$XY^+ + e^- = X + Y \qquad (1)$$

Loss rate of ion is $\alpha[XY^+][e^-] = \alpha N_e^2$, where $[\ ]$ denotes density, and $\alpha$ is the recombination rate coefficient. If the production rate of ion is equal to the loss rate of ion, we obtain

$$\alpha N_e^2 = P_{euv} + P_e \qquad (2)$$

This is Eq. (2) of Zhang et al. (2010). In Zhang et al. (2010), they describe

$$\alpha(N_e^{euv})^2 = P_{euv} \qquad (3)$$

$$\alpha(N_e^e)^2 = P_e \qquad (4)$$

where $N_e^{euv}$ represents the electron density due to solar EUV only and $N_e^e$ due to electron precipitation only. Then, they obtain

$$N_e = \sqrt{(N_e^{euv})^2 + (N_e^e)^2} \qquad (5)$$

Although the authors have followed Zhang et al. (2010), this reviewer considers that Eqs. (3)-(4) are incorrect, and that thus, Eq. (5) is also incorrect. The reason is described below.

1).

To obtain Eq. (3), this reviewer considers that Zhang et al. (2010) assume the following reaction for loss of the ion produced by solar EUV only

$$XY^{+\,euv} + e^{-\,euv} = X + Y \qquad (6)$$

where $XY^{+\,euv}$ and $e^{-\,euv}$ are the ion produced by solar EUV only, and electron precipitation only, respectively. They consider that loss rate of $XY^{+\,euv}$ is

$$\alpha[XY^{+\,euv}][e^{-\,euv}] = \alpha(N_e^{euv})^2 \qquad (7)$$

However, this reviewer doubts the reaction (6). Why does $XY^{+\,euv}$ react with only $e^{-\,euv}$? Why doesn't $XY^{+\,euv}$ react with $e^-$? How does $XY^{+\,euv}$ distinguish $e^{-\,euv}$ from $e^-$? Correct chemical reaction (6) should be

$$XY^{+\,euv} + e^- = X + Y \qquad (8)$$

because $XY^{+\,euv}$ reacts not only with $e^{-\,euv}$ but also $e^{-\,e}$. So, the loss rate of $XY^{+\,euv}$ is written as

$$\alpha[XY^{+\,euv}]\{[\,e^{-\,euv}] + [\,e^{-\,e}]\} = \alpha[XY^{+\,euv}][\,e^{-}] = \alpha N_e^{euv} N_e \qquad (9)$$

In the same way, the loss rate of $XY^{+\,e}$, which is the ion produced by electron precipitation only is written as

$$\alpha[XY^{+\,e}]\{[\,e^{-\,euv}] + [\,e^{-\,e}]\} = \alpha[XY^{+\,e}][\,e^{-}] = \alpha N_e^{e} N_e \qquad (10)$$

From Eq. (9)-(10), total loss of the ion is obtained as

$$\alpha(N_e^{euv} N_e + N_e^{e} N_e) = \alpha(N_e^{euv} + N_e^{e})N_e = \alpha N_e^{2} \qquad (11)$$

where

$$N_e^{euv} + N_e^{e} = N_e \qquad (12)$$

is used. Eq. (12) is consistent with loss rate shown in Eq. (2). Consequently, Eqs. (3)-(5) are incorrect.

2). By defining a percentage of $N_e^{e}$ in $N_e$ as $\gamma$,

$$N_e^{e} = \gamma N_e \qquad (13)$$
$$N_e^{euv} = (1-\gamma)N_e \qquad (14)$$

are obtained. Substituting Eq. (13) and (14) to Eq. (5),

$$N_e = \sqrt{(N_e^{euv})^2 + (N_e^{e})^2}$$
$$= \sqrt{(1-\gamma)^2 + \gamma^2}\,N_e$$
$$= \sqrt{2\gamma^2 - 2\gamma + 1}\,N_e$$

Thus, $1 = 2\gamma^2 - 2\gamma + 1$, $\quad \gamma(\gamma - 1) = 0$

$$\gamma = 0, 1$$

For the case of $\gamma = 0$, $N_e^{e} = 0$ and $N_e^{euv} = N_e$.

For the case of $\gamma = 1$, $N_e^{e} = N_e$ and $N_e^{euv} = 0$.

Eq. (5) is valid only when $N_e = N_e^{euv}$ or $N_e^{e}$. This result indicates that Eq. (5) is incorrect. This is because Eqs. (3)-(4) are incorrect.

In this manuscript, the authors follow Eqs. (3)-(4). This reviewer recommends the authors not to follow Eq. (3)-(5) of Zhang et al. (2010).

---

## Author Comment (AC2) · 19 Nov 2019

I think that you have misunderstood the meaning of $N_e^{euv}$ and $N_e^e$ in Zhang et al. (2010). These are quantities computed based solely on $P_{euv}$ and $P_e$, respectively. If you read the lines immediately following Zhang et al. eq. (5), you will see that the purpose of eq. (5) is to show how to combine electron densities from two different models, one providing $N_e^{euv}$ and the other providing $N_e^e$. (Actually, they provide NmE, but it is the same principle.) You write "$XY^{+euv}$ reacts not only with $e^{-euv}$ but also $e^{-e}$", but this is incorrect given the context, since Zhang et al. eq. (3) is for a physical setting in which ionization only results from solar EUV, and Zhang et al. eq. (4) is for a physical setting in which ionization only results from electron precipitation.

---

## Referee Comment (RC3) · Anonymous Referee #1 · 21 Nov 2019

Thank you for your explanation. I had misunderstood Zhang et al. eq. (3)-(5). I understand the author's explanation.

---

## Referee Comment (RC4) · Anonymous Referee #2 · 14 Mar 2020

This study focuses on effects of the proton precipitation on auroral E-region signatures. In reality, auroral particle precipitations include both electrons and protons. However, most previous studies on this topic used to assume pure electron precipitations except for some specific events. From this view point, it is considered that issue of the proton effect is important.

This article, however, is hard to follow its logic. One of the conclusions is summarized at lines 224-227 (at the top of Section 4.1). Based on results shown in Figure 1a, these sentences state that "it is expected that FUV-derived auroral NmE (i.e., derived under

the assumption of pure electron aurora) will be too high for pure proton aurora by a factor of ∼1.22". This is the case for kappa = 3.1, and for the other two cases kappa = 6.2 and 100, this conclusion cannot meet individual results because the estimated values are in a same level as the estimated for the pure electron case. The article should explain the reason to focus on the case of kappa = 3.1 alone in more detailed. Furthermore, it is unclear for me why NmE is the appropriate parameter to evaluate the proton/electron contributions. The article should mention this point clearly.

Discussion related to the LBHL emission is also unclear for me. Line 148 tells that "... LBHL values are ∼50% higher than for electron spectra with the same precipitating energy flux and LBHS/LBHL values." According to Figure 1c, this is the case for LBHS/LBHL from about 0.5-0.8. For the LBHS/LBHL outside of this range, this is not the case or even LBHL intensity for the pure electron case can be higher than that for the proton cases. Discussion written at Lines 215-216 has been developed taking into account "50% higher" case alone, and there is no consideration on ambiguities of the ratio of the proton case to the electron. Since a part of conclusions in this study has been made by discussion at Lines 215-216, that no consideration is serious lack for making the conclusion.

---

## Author Comment (AC3) · 17 Mar 2020

Reviewer comment:

This article, however, is hard to follow its logic.

HK reply:

I am sorry that you have this impression of the paper. The paper is considerably shorter than my previous paper on FUV-ionosonde comparisons, Knight et al. (2018), so I was

hoping that readers would find it less challenging.

I was also first author on a related paper, Knight et al. (2012), "An empirical determination of proton auroral far ultraviolet emission efficiencies using a new non-climatological proton flux extrapolation method". In my opinion, Knight et al. (2012) was considerably more challenging for readers than the current paper.

I am willing to make limited changes to the current paper to clarify the specific issues you have raised.

Reviewer comment:

One of the conclusions is summarized at lines 224-227 (at the top of Section 4.1). Based on results shown in Figure 1a, these sentences state that "it is expected that FUV-derived auroral NmE (i.e., derived under the assumption of pure electron aurora) will be too high for pure proton aurora by a factor of ∼1.22". This is the case for kappa = 3.1, and for the other two cases kappa = 6.2 and 100, this conclusion cannot meet individual results because the estimated values are in a same level as the estimated for the pure electron case. The article should explain the reason to focus on the case of kappa = 3.1 alone in more detailed.

HK reply:

The factor of 1.22 is explained in section 3 at lines 148-151. The factor of 1.22 is obtained as the square root of 1.5, where 1.5 is obtained as the factor resulting from LBHL values being ∼50% higher for electron spectra than for proton spectra for the same precipitating energy flux. Note that 50% is equivalent to a factor of 1.5 and that this is where the factor of 1.5 comes from. I can add a statement to this paragraph to clarify this particular logical connection.

In lines 151-153, it mentions that the situation is more complicated for kappa=3.1 but that there will still be an algorithm bias of 1.22 for kappa=3.1. In other words, the factor of 1.22 applies to all kappa values. You concluded that the factor of 1.22 was specific

to kappa=3.1, but it was actually explained in the lines 148-151 that the factor of 1.22 applies to the other proton spectral shapes as well.

When you write "This is the case for kappa = 3.1, and for the other two cases kappa = 6.2 and 100, this conclusion cannot meet individual results because the estimated values are in a same level as the estimated for the pure electron case", this reflects a misperception on your part. You are basing this statement on Figure 1a, but actually the factor of 1.22 is explained at lines 148-153, as just described.

Reviewer comment:

Furthermore, it is unclear for me why NmE is the appropriate parameter to evaluate the proton/electron contributions. The article should mention this point clearly.

HK reply:

The main motivation for the paper is that a bias in FUV-derived NmE (meaning NmE predicted by auroral FUV remote sensing algorithms) is expected in the presence of proton precipitation. As explained at lines 213-218, if proton aurora is a factor of 2 more efficient in producing LBH than electron aurora (as predicted by previous papers), then it is expected that FUV-derived NmE will be too high by a factor of 1.73. This means that it is expected that auroral FUV remote sensing algorithms will report NmE values that are too high by a factor of ∼1.7 for proton aurora.

There are several different effects involved here, including emission efficiencies, model-predicted NmE values, and the way auroral FUV remote sensing algorithms work. I think that there is enough information in the paper to allow readers to understand the issues, but in order to help readers, I could add a parenthetical statement after line 218 to remind readers of why LBH biases imply NmE biases in auroral FUV remote sensing algorithms. It has to do with NmE being approximately proportional to the square root of Q.

Reviewer comment:

Line 148 tells that "... LBHL values are ∼50% higher than for electron spectra with the same precipitating energy flux and LBHS/LBHL values." According to Figure 1c, this is the case for LBHS/LBHL from about 0.5-0.8. For the LBHS/LBHL outside of this range, this is not the case or even LBHL intensity for the pure electron case can be higher than that for the proton cases. Discussion written at Lines 215-216 has been developed taking into account "50% higher" case alone, and there is no consideration on ambiguities of the ratio of the proton case to the electron. Since a part of conclusions in this study has been made by discussion at Lines 215-216, that no consideration is serious lack for making the conclusion.

HK reply:

First of all, one cannot tell directly from Figure 1c what the expected bias is based on the observed LBHS/LBHL ratio for any particular example. The purple curve in Figure 1c is for pure electron aurora, and blue, green and orange curves are for pure proton aurora. There are no LBHS/LBHL ratios outside of ∼0.4 to ∼0.9 (according to the model) for pure proton aurora. Suppose that for a particular example of observed LBHS and LBHL there is a mix of electron and proton aurora, with the portion of LBH due to electron precipitation having an LBHS/LBHL ratio of 0.2 and the portion due to proton precipitation having an LBHS/LBHL ratio of 0.6. Depending on the relative energy fluxes of electron and proton aurora, the observed LBHS/LBHL ratio (setting aside observation error) could be anywhere between 0.2 and 0.6. Suppose that the observed LBHS/LBHL ratio is 0.4, near where the orange curve crosses the purple curve. This does not mean that the precipitating proton and electron spectra have the same LBHL yields. The proton LBHL yield is likely to be around 270-300 R/(ergs/cm^2/s), while the electron LBHL yield will be around 140 R/(ergs/cm^2/s) (based on the assumed LBHS/LBHL ratios mentioned above for this particular example).

Having said that, there is still an underlying issue of the variation of LBHL yields with precipitating particle and spectra types and whether the predicted biases in auroral FUV remote sensing algorithms are realistic given the aforementioned variation. The

main way I have addressed this in the paper is by giving results of a simulation in the appendix. The simulation is based on an entire year of TED and MEPED observations of precipitating electron and proton fluxes. The observations were filtered (in the simulation) in such a way as to give spectra that would be representative of the spectra occurring in the actual coincident FUV-ionosonde observations.

To further address the issue of variation of spectral types, I could add a statement such as the following to the main text: "Based on an examination of in situ particle flux data, we have seen that in cases where proton Q is comparable to or greater than electron Q, proton aurora typically has a ∼50% greater model-predicted yield than electron aurora. Details of such analysis are omitted, but the results of related analysis based on in situ electron and proton flux observations are given in the appendix."

Thank you for your comments.

---

## Author Comment (AC4) · 1 Apr 2020

I did not realize that a reply was needed. The reviewer wrote "Thank you for your explanation. I had misunderstood Zhang et al. eq. (3)-(5). I understand the author's explanation." This shows that the matter is settled.

---

## Author Response (AR1)

**Please see the end of this document for an explanation of the revisions.**

[revised manuscript text omitted]

**Explanation of revisions:**

Here I will explain the reasons for changes made to the manuscript, including how the changes respond to reviewer
comments. I will not try to repeat all of the explanations that I gave during the interactive discussion phase. If a reviewer thinks that a previously raised issue has not been adequately addressed here or in the revisions, please also see my comments during the interactive discussion phase.

The line numbers given below refer to the marked up text showing changes above. Most of the changes are in response to
reviewer comments, but a few were made to improve wording and/or clarity or to fix typographical errors. Blue font will be used for quotes from reviewer comments.

Line 29: Fixed a typographical error.

Line 119: Improved wording.

Lines 137-139: This improves clarity by defining Q and partly responds to RC4 comment "According to Figure 1c, this is the case for LBHS/LBHL from about 0.5-0.8. For the LBHS/LBHL outside of this range …" For pure proton aurora, ratios outside of the approximate range you indicated do not occur generally. Ratios outside of this range only occur for pure
electron and mixed aurora. Further revisions discussed below also respond to this.

Line 149: This responds to the RC1 comment "Does "local equilibrium" means the equilibrium between the ionization and recombination rates of plasma in the E region? If so, it is better to describe so."

Lines 151-156: This responds to the RC1 comment "This explanation seems strange. If production rates of the ion by electron and proton aurora…" It also partly responds to the RC4 comment "Furthermore, it is unclear for me why NmE is the appropriate parameter to evaluate the proton/electron contributions" which will also be addressed by further revisions discussed below.

Line 160: This partly responds to the RC4 comment "This is the case for kappa = 3.1, and for the other two cases kappa = 6.2 and 100, this conclusion cannot meet individual results because the estimated values are in a same level as the estimated for the pure electron case." I wanted to make the explanation for the 1.22 factor (obtained as the square root of the 1.5 factor introduced at line 160) clearer, since the reviewer's statement reflects a misunderstanding. This is also addressed below.

Lines 163-164: This responds to the same RC4 comment mentioned above for line 158. It shows how 1.22 is obtained from the 1.5 factor mentioned above and restates the meaning of the 1.5 factor.

Lines 164-171: Here I have added a few more lines clarifying the issue raised by the RC4 comment mentioned above for line 160. What the reviewer missed is that the 1.22 factor applies to the two narrow proton spectra, and the kappa=3.1 case (and,
by extension, other wide proton spectral shapes) have to be considered separately, which is done in the new lines.

Lines 172-177: This further responds to the RC4 comment "According to Figure 1c, this is the case for LBHS/LBHL from about 0.5-0.8. For the LBHS/LBHL outside of this range …". I responded in detail to this comment during the interactive discussion phase and stated that I would add statements to the manuscript clarifying the issue of variation in spectral shapes in mixed aurora, which is the purpose of these new lines.

Lines 240-244: This responds to the statement "Furthermore, it is unclear for me why NmE is the appropriate parameter to evaluate the proton/electron contributions. The article should mention this point clearly." In the auroral FUV remote sensing algorithm, bias in derived NmE will be proportional to the square root of the bias in LBH.

Line 362: This was to improve wording.

Line 371: This fixes a typographical error.

Line 374: This improves clarity.

Lines 378-383: These changes improve clarity.

Lines 392 and 395: These changes improve wording and clarity.

Line 416: This change improves clarity.

Lines 614-615: This reference is added in response to the RC1 comments.

---

## Referee Report (RR1)

[referee-annotated manuscript omitted]

---

## Author Response (AR2)

**Please see the end of this document for the response to the reviewer's comments.**

[revised manuscript text omitted]

$$\text{GUVI}_i(\text{Res(NmE)}, \text{pre}) = \frac{\text{GUVI}_i(\text{NmE,I,pre})}{\exp(\text{GUVI}(\alpha,\text{pre})+\text{GUVI}(\beta,\text{pre})\ln(\text{GUVI}_i(\text{NmE,F})))}. \tag{6}$$

This generalized type of residual, which will sometimes be referred to simply as Res(NmE), is dimensionless, since the right side of eq. (6) has a ratio of NmE values. Although "residuals" usually refer to differences and not ratios, note that a ratio can be interpreted as a difference in logarithmic space.

310  When data values for all three FUV instruments are combined, they will be referred to with the same notation as before, except with the instrument abbreviation replaced by "COMB." The calculations of residuals and PIVs are still instrument-specific and are performed prior to combining the derived values. Let $\text{COMB}(\text{Res(NmE)}, \text{PIV} < 0.5, \text{pre})$ and $\text{COMB}(\text{Res(NmE)}, \text{PIV} \geq 0.5, \text{pre})$ be the subsets of $\text{COMB}(\text{Res(NmE)}, \text{pre})$ for which PIV is or less than 0.5 and greater than or equal to 0.5, respectively. These will be referred to as "below" and "above" sets. Similar definitions can be made for

315  the post-coincidence ionograms and the individual FUV instruments. The statistical effect of proton aurora on Res(NmE) will now be described in terms of the ratio of medians

$$\text{COMB}(\text{BAR}, \text{pre}) = \frac{\text{median}\left(\text{COMB}(\text{Res(NmE)},\text{PIV}<0.5,\text{pre})\right)}{\text{median}\left(\text{COMB}(\text{Res(NmE)},\text{PIV}\geq0.5,\text{pre})\right)}, \
[revised manuscript text omitted]

**Response to reviewer:**

All line numbers refer to the marked up version above.

Thanks for reviewing my paper.

660

Please see the attached commented pdf of the original manuscript for reference. My ultimate recommendation is for the manuscript to be accepted pending minor revisions. A summary of my review can be found below:

665 1) Overall, this study is a rather incremental advance with respect to the author's previous work on the topic, using the same ionosonde data, but nonetheless, I believe the results warrant publication.

I would like to see some commentary on the expected implications of these results in the context of models based on these FUV observations, such as the Zhang and Paxton [2008] model and their

subsequent suggestion (and work) of including such a model in the International Reference Ionosphere (IRI).

https://doi.org/10.1016/j.jastp.2008.03.008

I have added a paragraph at the end of the discussion and conclusion section (lines 473-480) on this issue you raised.

2) The Poker Flat ISR has been operating nearly continuously since 2007, almost all of which with a high resolution alternative code mode that gets sufficient resolution in the E-Region for your purposes. Given the substantial challenges and data quality concerns that may arise using the ionosonde data, I am a bit disappointed that there was no effort to use the ISR data, especially given that this would alleviate the need for the manual interpretation of ionosonde data, which is somewhat prohibitive. I imagine a future study may do so, but that sounds like an adventure in stamp collecting rather than scientific progress...

Thanks for the suggestion. My impression (which may be out of date) has been that suitable ISR observations of the auroral E-region are scarce relative to ionosonde observations. If I missed a plentiful source of FUV-coincident ISR observations, that is unfortunate. I prefer not to write anything about Poker Flat ISR in this paper, since I am not sufficiently familiar with that source of E-region data.

3) Given the null result of your hypothesis, why have you not examined potential limitations in the manner in which NmE is being derived from the LBHS/LBHL data? Could the proton effect actually be there, but be countered by an issue in the FUV-derivation of NmE itself? I believe that shortcomings in the NmE estimation must be addressed in order to cover the loose ends.

The level of consistency between FUV-derived NmE and ionosonde-observed NmE was studied in depth in Knight et al. (2018). Section 4.2 of the current paper considers a number of possible types of explanations for the discrepancy. I could not find any plausible explanation that would account for the statistical agreement between FUV-derived and ionsonde-observed NmE found in Knight et al. (2018) while also accounting for the missing expected discrepancy in the current paper. Please see section 4.2 in the current paper, with the parts most directly relevant to the issue you raise beginning at line 424 (marked-up version).

4) Regarding Figure 2: The statistical spread about these fits is on the order of the span on the data. More samples are likely necessary in order to really draw conclusions from this plot...

Conclusions are not drawn directly from Figure 2. I added a paragraph at lines 291-297 to give more context. The purpose of the figure is to illustrate what is being done. Conclusions are drawn from Figure 3 and related results discussed in section 4.1 and appendix A.

5) Lines 435-436: Please provide a reference.

Done. The line number in the marked-up version is 444.

715 6) Please examine the GIRO rules of the road document and include the necessary acknowledgments for ionosonde operators that contribute to GIRO: https://ulcar.uml.edu/DIDBase/Acknowledgements.htm

Best practices are to contact operators about acknowledgment if they are not listed in the above list. For example, Tromso requires that authors acknowledge the following: "Tromsø ionosonde operations are
720 funded in part by QinetiQ"

I contacted the operators and added more specific acknowledgement statements for the four stations.